# Free-Range and Low-Protein Concentrated Diets in Iberian Pigs: Effect on Plasma Insulin and Leptin Concentration, Lipogenic Enzyme Activity, and Fatty Acid Composition of Adipose Tissue

**DOI:** 10.3390/ani10101917

**Published:** 2020-10-19

**Authors:** Juan F. Tejeda, Alejandro Hernández-Matamoros, Elena González

**Affiliations:** 1Food Science and Technology, Escuela de Ingenierías Agrarias, Universidad de Extremadura, Avda. Adolfo Suárez s/n, 06007 Badajoz, Spain; alejandro@nutricionespecial.com; 2Research University Institute of Agricultural Resources (INURA), Avda. de Elvas s/n, Campus Universitario, 06006 Badajoz, Spain; malena@unex.es; 3Animal Production, Escuela de Ingenierías Agrarias, Universidad de Extremadura, Avda. Adolfo Suárez, 06007 Badajoz, Spain

**Keywords:** Iberian pig, extensive system, low-protein diet, adipose tissue, plasma hormones, lipogenic enzymes, fatty acids

## Abstract

**Simple Summary:**

Recently, it has been shown that reducing dietary crude-protein levels during the final fattening period prior to slaughter is a suitable strategy to increase intramuscular fat content in Iberian pig meat, without affecting pig growth. Investigating the effect of a low-protein diet on the metabolism, development, and composition of the adipose tissue of Iberian pigs, and the obese porcine breed, was the objective of this study. Three groups of pigs fed under free-range conditions and in confinement with concentrated diets with low- and standard-protein contents were studied. All three groups exhibited the same backfat thickness at the end of the fattening period. The level of hormones and activities of enzymes related to adipogenic metabolism were affected by diet, with differences between free-range and intensive feeding systems. Therefore, we suggest that feeding Iberian pigs on low-protein diets did not result in fatter carcasses, and is thus a useful strategy to improve Iberian pig meat quality.

**Abstract:**

The purpose of this study was to investigate the effect of diets with different protein contents on carcass traits, plasma hormone concentration, lipogenic enzyme activities, and fatty acid (FA) composition in the adipose tissue of Iberian pigs. Twenty-four castrated male Iberian pigs (eight per feeding diet) were fed under free-range conditions with acorns and grass (FR), and in confinement with concentrated diets with standard (SP) and low-protein contents (LP) from 116.0 to 174.2 kg live weight. Backfat thickness was not affected by diet. The plasma leptin concentration was higher (*p* < 0.001) in the FR group than in the LP and SP groups, while insulin concentration was higher in the SP group than in the LP and FR groups. The lipogenic enzyme activities of glucose-6-phosphate dehydrogenase, malic enzyme, and glycerol-3-phosphate dehydrogenase were lower in the FR group compared to the LP and SP pigs. The activities of these enzymes were adipose-tissue-specific. No differences were found in FA composition of adipose tissue between the SP and LP groups, while the FR pigs had lower proportions of saturated FA and higher proportions of monounsaturated and polyunsaturated FA than the SP and LP pigs. In conclusion, feeding low-protein diets in Iberian pigs does not seem to affect adipose carcass traits, strengthening previous findings that indicate that this is a good strategy to improve meat and dry-cured product quality.

## 1. Introduction

The high level of crude protein in pig diets, usually formulated to provide the requirements for certain essential amino acids, leads to an excess of other essential amino acids [1] and excretion of excess nitrogen [2], resulting in a lower efficiency of nitrogen utilization, and also protein fermentation in the hindgut, which can damage the gut health [3]. Moreover, in the last years, there has been increased interest in reducing ammonia emissions from pig farms due to their negative impact on the environment and on human health [4]. Therefore, feeding pigs on low-protein diets is a tool not only to reduce feed costs, nitrogen excretion, and gut health, but also could contribute to improving meat quality without affecting pig production performance. In this sense, several studies have been carried out to explore the effect of different nutritional strategies, based on reducing crude-protein content [5,6,7,8,9,10], essential amino acids levels, such as lysine [11,12,13,14,15], or both of these [16] in the diet, on growth performance, carcass composition, and meat quality of pigs. It is well known that a high amount of intramuscular fat (IMF) is one the most relevant aspects of meat quality [17], and although low-fat pork could be interesting for reducing caloric intake in humans, an IMF level below 2.5% is related to lower sensory meat quality [18]. In a recent paper, we evidenced that reduced dietary crude-protein levels during the final fattening period prior to slaughter in intensive conditions, as occur in free-range pigs reared with natural resources (acorns and grass) in the south-west Iberian Peninsula, is a suitable strategy to increase IMF content in Iberian pig meat, without affecting the pig growth [19]. The Iberian pig is an autochthonous breed characterized by its low potential for lean tissue deposition and its high tendency to accumulate fat [20], as a consequence of an excessive and continuous intake, since their sensation of satiety is altered [21]. Therefore, it is of great importance to understand how a low-protein diet affects the biochemical mechanisms related to the production and development of adipose tissue. In this sense, some authors have reported that protein-deficient diets increased backfat thickness [5,22] or modified monounsaturated fatty acids (MUFA) biosynthesis in muscle, by increasing the activity and protein expression of stearoyl-CoA desaturase [23]. Katsumata et al. [11] reported an increase in m(messenger)RNA abundance of PPARG (the nuclear hormone receptor involved in adipocyte gene expression and differentiation) in the *longissimus* muscle. However, the effects of this type of diet on biochemical mechanisms are different in muscle and subcutaneous backfat-adipose tissue [23]. In this context, it is very interesting to determine the effect of the reduction in the protein content in the pig diet, not only on muscle tissue, but also on the metabolism, development, and composition of the adipose tissue, mainly on the backfat, especially in the case of fatty pigs, as is the case with the Iberian pig. The present study is, to our knowledge, the first report to look at the hormone profile and lipogenic enzyme activity of the Iberian pig fed in free-range conditions compared with pigs fed intensively on concentrate diets. With this background, we hypothesized that exploring the effect of dietary protein deficiency on metabolic changes, studying plasma hormones and lipogenic enzyme activity, and the fatty acid composition of different adipose tissues from Iberian pigs fed in free-range and intensive conditions could help better understand the findings previously reported [19] on the effect of this feeding strategy on Iberian pig productive performances and meat quality.

## 2. Materials and Methods

### 2.1. Animals, Experimental Design, and Diets

The experimental design carried out in this study was described in detail in a previous paper [19]. Briefly, 24 castrated male Iberian pigs of the Retinto variety were selected at a 116.0 ± 5.9 kg live weight and an age of 425 days, and randomly divided into 3 groups according to the rearing system during the fattening period. Eight pigs were fattened in the traditional extensive rearing system based on local resources (acorn and grass) in a large enclosure (FR). The others were fed on two different concentrate diets: 1 group of 8 pigs was fed on a standard protein diet (SP) and the other on a low-protein diet (LP); both groups were fed in confinement. Diet composition, including daily feed intake of acorn, grass, and concentrates; handling; and carcass sampling were detailed in the above-mentioned study by Tejeda et al. [19]. In brief, pigs were fattened for 45–57 days (45.4, 46.3, and 56.8 days for LP, SP, and FR pigs, respectively) in order to reach a similar slaughter weight (174.2 ± 6.1 kg live weight), and daily feed intakes were of 5161 and 5088 g DM/pig in LP and SP pigs; while in FR pigs estimated daily feed intakes were of 3600 and 655 g DM/pig for acorn pulp and grass, respectively. The backfat thickness of pigs was measured using a hand-held ultrasonic device (Aquila vet, Esaote Pie Medical, Genoa, Italy), equipped with an ultrasonic linear probe (3.5 Mhz and 18 cm long) with a silicone acoustic loin adapter, by placing the probe perpendicular to the loin at the level of the last rib. The captured images were processed using the AutoCAD^®^ 2008 software (AutoDESK^®^, Inc., San Rafael, CA, USA) to determine total backfat thickness and the thickness of the 3 layers that can be differentiated into subcutaneous fat [24] at 10 cm from the dorsal midline. These 3 layers will be referred to as SC1, SC2, and SC3, for each of the outer, middle, and inner layers, respectively, in the following text. Pigs were restrained in a crate during ultrasound scanning to restrict movement and maintain a standing posture and were ultrasonically scanned the on the first and the last day (24 h before slaughter) of the fattening period. The backfat thickness increase was calculated as the difference between the 2 ultrasonic measurements, and divided by the total number of days in fattening period, and multiplied by 10 (backfat thickness increasing every 10 days). Feed was withheld from animals for 12 h before slaughter. FR pigs were locked in a pen and fed with acorns collected from the same place where they usually ate, until the pre-slaughter fasting. Pigs were slaughtered by electrical stunning and killed by exsanguination. Then, they were scalded, skinned, eviscerated, and split down both sides of the vertebral column according to the standard commercial procedures of the Iberian pig industry. Hot carcass weights without pelvic renal fat were recorded and used to calculate carcass yield. Samples of perirenal fat (PE), backfat, at the level of the tailbone (TB), and samples from SC1, SC2, and SC3 layers of backfat at the level of the last rib were collected for lipid analysis, frozen in liquid nitrogen, and stored at −80 °C until analysis. Blood samples were collected during postmortem exsanguinations and were immediately stored at 4 °C in tubes with heparin to prevent clotting, until arrival at the laboratory. In the laboratory, blood samples were centrifuged at 5000× *g* for 10 min, and plasma was collected and stored at −80 °C until analysis. Hams and shoulders were removed from the carcasses, weighed 2 h postmortem, and trimmed 24 h later according to the approved procedure for the production of Iberian ham; they were weighed again to calculate trimmed yields.

### 2.2. Analyses

#### 2.2.1. Assessment of Plasma Hormone Concentrations

The circulating leptin and insulin concentrations were measured using the radioimmunoassay (RIA) kits Multi-Species Leptin RIA Kit (cat. no. XL-85K, Merck Millipore, Darmstadt, Germany) and Porcine Insulin RIA Kit (cat. no. PI-12K, Merck Millipore, Darmstadt, Germany), respectively. The leptin and insulin assays were performed according to the manufacturer’s instructions, including standard and sample tubes. Briefly, for leptin determination, 300 μL of assay buffer, 100 μL of standards, and 100 μL of each sample were added to the respective tubes, followed by 100 μL of Multi Species Leptin antibody; they were then vortexed, covered, and incubated overnight at 4 °C. After incubation, 100 μL of 125I-Human Leptin was added and incubated overnight again at 4 °C. On the third day, 1.0 mL of cold (4 °C) precipitating reagent was added to the tubes, vortexed, and incubated for 20 min at 4 °C. After that, the tubes were centrifuged for 20 min at 2500× *g*, the supernatant was decanted and counted in a gamma counter (model 5500; Beckman Instrument, Irvine, CA, USA) for 1 min. The analyses were carried out in duplicate, and the results were expressed as ng/mL human equivalent of leptin. For insulin determination, after adding the assay buffer (300 μL), standards (100 μL), and samples (100 μL) to the tubes, 100 μL of 125I-Insulin and 100 μL of Porcine Insulin antibody were added. After mixing, this was incubated for 24 h at 4 °C. On the second day, 1.0 mL of cold (4 °C) precipitating reagent was added to the tubes and re-incubated for 20 min at 4 °C. Finally, the tubes were centrifuged (20 min at 2500× *g*) and the supernatant was decanted and counted in the gamma counter for 1 min. The analyses were carried out in duplicate, and the results were expressed as μU/mL of porcine insulin. In both the leptin and insulin assays, the difference between the duplicate results of a sample was <10% coefficient of variation, and the sensitivity was <1 ng/mL and <2 μU/mL, respectively.

#### 2.2.2. Assessment of Lipogenic Enzyme Activity

The activities of 4 lipogenic enzymes, glucose-6-phosphate dehydrogenase (G6PDH) (EC 1.1.1.49) [25], malic enzyme (ME) (EC 1.1.1.40) [26], glycerol-3-phosphate dehydrogenase (G3PDH) (EC 1.1.1.8) [27], and fatty acid synthetase (FAS) (EC 2.3.1.85) [28] were assessed on subcutaneous (TB, SC1, SC2, and SC3) and perirenal adipose tissue homogenates. The analytical procedure followed was: 0.7 g of adipose tissue, previously frozen at −80 °C, was homogenized in 4 mL of STEG ice-cold buffer (containing 300 mM saccharose, 30 mM trizma base, 1 mM EDTA and 1 mM glutatión, at pH = 7.4) using an Omni-Mixer homogenizer (OMNI Int., Waterbury, CT, USA) at 50,000 rpm for 3 cycles of 10 s each. The homogenate was filtered (20 μm pore size filter) and centrifuged at 4000× *g* for 10 min at 4 °C in an Eppendorf Centrifuge 5810R (Eppendorf, Hamburg, Germany). The supernatant fraction was re-filtered (0.45 µm pore size filter) and re-centrifuged at 14,000× *g* for 10 min at 4 °C. Then, the resulting supernatant (cytoplasmic soluble fraction) was collected, filtered (0.45 µm pore size cellulose filter), and stored at −80 °C for further analyses. The G6DPH, ME, G3PDH, and FAS activities were assessed at 37 °C by absorbance at 340 nm using a spectrophometer (Fluostar Optima, BMG Labtech, Aylesbury, UK). One unit of activity was defined as the amount of enzyme that increases (for ME and G6PDH) or decreases (for FAS and G3PDH) the presence of 1 nmol of nicotinamide adenine dinucleotide phosphate (NADPH) (NADH for G3PDH) per minute and per gram of fresh tissue. The amount of substrate was adjusted so that the reactions were linear over time during the assay period.

#### 2.2.3. Fatty Acid Composition

Fatty acid compositions of backfat and perirenal fat samples were determined, after lipid extraction in a microwave oven [29], by acidic trans-esterification with 0.1 N sodium metal and 5% sulphuric acid in methanol [30]. The fatty acid methyl esters were analyzed using a Hewlett-Packard HP-4890 Series II gas chromatograph (Hewlett-Packard, Palo Alto, CA, USA) equipped with a split/splitless injector and a flame ionization detector (FID). The derivatives were separated on a capillary column (HP-INNOWax 30 m long, 0.25 mm id, 0.25 µm film thickness; Hewlett-Packard, Palo Alto, CA, USA) containing a polar stationary phase (polyethylene glycol). The oven temperature was held at 260 °C for 25 min. The injector and detector temperatures were held at 320 °C. The carrier gas was nitrogen at 1.8 mL/min. The methyl esters were identified by comparing their retention times with those of the reference standard mixtures (Sigma Chemical Co., St. Louis, MO, USA). The results were labeled as a percentage of the total fatty acids present, considering a total of 15 fatty acids.

### 2.3. Statistical Analysis

The mean and the standard error of the mean were used for the descriptive data analysis. The pig was used as the experimental unit. One-way ANOVA was used to determine the effect of dietary treatment on carcass traits and plasma hormone concentrations. The effect on lipogenic enzyme activities and fatty acid compositions of the backfat and perirenal fat, and their interaction, was carried out by Factorial (3 diet × 5 adipose tissue) ANOVA procedure. Tukey’s test was applied to compare the means of each group. The General Linear Model procedure of the SPSS package (SPSS for Windows Ver. 19.0; SPSS Inc., Chicago, IL, USA, 2004) was used. Differences between means were significant for *p* < 0.05.

## 3. Results

### 3.1. Carcass Measurements and Plasma Hormones

The results from the trimmed ham and shoulder, and backfat traits and from the plasma hormone concentration are shown in Table 1. According to the data published by the authors in a previous paper [19], the trimming, carried out according to the procedure approved for the production of Iberian ham, leads to a reduction in the weight of the ham and shoulder by 21% and 30%, respectively. The effect of the feeding system was only significant (*p* < 0.05) for trimmed shoulder yield, with the SP group showing higher scores than the FR and LP groups. Backfat thickness, at the level of the last rib, varied between 58.0 and 62.8 mm, without significant differences between the different feeding systems. With respect to the thickness increase in the three backfat layers studied, it was observed that the middle layer (SC2) showed the highest increase during the pig fattening period, followed by the inner layer (SC3), and finally the outer one (SC1). However, no significant differences were observed related to diet in terms of increasing thickness of any of the three layers studied. Regarding hormone concentrations, leptin contents varied widely, with significantly higher levels (*p* = 0.001) in the FR group than in the LP group; the SP pigs showed the lowest concentration. In contrast, plasma insulin concentration was higher (*p* = 0.007) in the SP group than in the LP group, with the FR animals showing intermediate levels.

### 3.2. Fat Biochemical Characteristics

The lipogenic enzyme activities of the four enzymes determined in the different diets and adipose tissues studied are shown in Table 2. The activities of the G6PDH and ME enzymes were similar (on average, 427 and 421 nmol NADPH/min/g of lipids, respectively), but higher than the G3PDH activity (189 nmol), with the FAS enzyme demonstrating the lowest activity (34 nmol). The activity of G6PDH, ME, and G3PDH was affected by the feeding treatment. The FR diet displayed significantly lower activities of G6PDH (−25%, *p* < 0.001), ME (−34%, *p* < 0.001), and G3PDH (−35%, *p* < 0.001) compared to the LP and SP diets. The FAS activity was not affected by diet.

For TB, backfat layers, and PE, significant differences were also detected in G6PDH, ME, and G3PDH enzyme activities. The FAS activity did not differ between the five different tissues studied. For G6PDH, TB exhibited the highest enzyme activity, followed by SC1 and SC2, SC3, and finally, PE. A similar trend was followed by G3PDH, with the highest levels being in TB and the lowest in PE and SC3; SC1 and SC2 showed intermediate levels. In contrast, ME showed lower enzyme activity in TB and PE than the SC layers. No interaction was observed between diet and tissue.

### 3.3. Fatty Acid Composition

The fatty acid composition of total subcutaneous fat at the level of the tail bone (TB), including the outer (SC1), middle (SC2), and inner (SC3) layers, and perirenal (PE) fat is presented in Table 3. Pigs reared in free-range conditions (FR) had significantly lower proportions (*p* < 0.01) of C16:0, C18:0, C20:0, and total saturated fatty acids (SFA) than pigs fed intensively with standard (SP) and low-protein diets (LP). In contrast, FR pigs showed higher (*p* < 0.05) percentages of C18:1 n-9, C18:2 n-6, C18:3 n-3, total monounsaturated (MUFA), and total polyunsaturated fatty acids (PUFA) than SP and LP pigs. However, no differences were found in any of the above-mentioned fatty acids between the SP and LP groups. With respect to the fatty acid composition of the different adipose tissues studied, the effect was significant in most of the fatty acids analyzed. Briefly, the fatty acid composition of total backfat at the level of the tail bone (TB) and the outer layer of subcutaneous fat (SC1) at level of the last rib were quite similar, but different to the SC2 and SC3 layers. Perirenal fat (PE) presented a different fatty acid composition, mainly SFA and MUFA, compared to the other fat tissues studied. The proportions of C18:1 n-9, C20:1 n-9, and total MUFA were higher in TB and SC1 than in SC2 and SC3, with PE showing the lowest percentages (*p* < 0.001). In contrast, total SFA, including C16:0 and C18:0, the two main saturated fatty acids, were lower in TB and SC1 compared to SC2 and SC3, with the highest proportions in PE (*p* < 0.001). Regarding PUFA, significant differences (*p* < 0.05) were observed between the different tissues studied, although they were less marked than those described in SFA and MUFA. No significant effect (*p* > 0.05) of interaction between the diet and adipose tissue was observed.

## 4. Discussion

The purpose of this study was to investigate the influence of a protein-restricted diet on the lipogenic response of the main fat depots of Iberian pigs carcasses, such as subcutaneous and perirenal fat [31]. Subcutaneous fat tissue in pigs consists of two or three layers (SC1, SC2, and SC3), depending on the carcass point [24], with differences in allometry coefficients and the composition between them [32], which suggests that it would be better to study the composition of individual fat layers as an indicator of carcass composition [33]. However, the Spanish Ministry of Agriculture, Fishery, and Food established a procedure to take backfat samples in Iberian pigs from the area of tail insertion in the coxal region of the carcass (TB), where no differentiation between the different layers is observed [34], in order to avoid the possible error that would be made if only one of the layers were sampled. Perirenal fat (PE) is a tissue that is found close to pig maturity [35], which probably could better reflect the diet composition at the end of the fattening period.

### 4.1. Carcass Traits

Trimmed shoulder yield was affected by Iberian pig diet, which implies a higher lean percentage in the SP trimmed shoulders than in the FR and LP trimmed shoulders, in agreement with the results observed previously in *longissimus thoracis* and *lumborum* muscle [19]. Trimming, the procedure that removes rind and external fat in the traditional V-shape, facilitates the salting phase and standardizes the subcutaneous fat thickness [36]. However, no significant effect of diet was observed as regards the final weight and yield of trimmed hams, in accordance with others studies carried out in Cinta Senese [10] and Large White × Landrace [7] pigs, respectively. This is important because ham is the main dry-cured product obtained from Iberian pigs, characterized by its high sensory quality and high price in the market [37]. In the same way, backfat thickness and backfat thickness increase were not affected by diet, showing that low-crude-protein diets did not result in fatter carcasses, in agreement with the data from previous studies in heavy [7,13] and lighter [8,38] pigs. Furthermore, other authors [32,39] reported that total backfat thickness and the different backfat layers thickness were no influenced by the feeding of Iberian pigs in extensive or intensive conditions. In the same way, these authors reported a slight increase over the finished fattening period in the external subcutaneous layer compared to the medium and internal ones, the medium layer representing nearly 60% of the total backfat [39]. A similar trend was observed by González et al. [40], who reported that the middle layer of backfat showed more growth than the inner layer, with the outer layer demonstrating the lowest fat thickness. In contrast, the increase in backfat thickness in pigs fed on low-protein diets has been previously reported [5]. These discrepancies between the published results could be related to the extent of crude-protein reduction in the diet, the amount of indispensable amino acids supplemented, and the net energy of the diet [7]. A great reduction in protein percentage, i.e., greater than about 3% below the control values, could significantly affect backfat thickness [8], since this diet protein reduction would cause levels of other important amino acids to be too low [41].

### 4.2. Plasma Hormones

Regarding the plasma hormone concentrations, significant differences in leptin and insulin contents were observed between the three diets studied. Iberian pigs fed on the SP diet had 55% and 35% less leptin content than the FR and LP groups, respectively. Leptin is a protein encoded by the gene responsible for obesity. The positive correlation between plasma leptin levels, pig fatness [21], and subcutaneous fat depth [42] is well documented, since this hormone is synthesized and secreted by adipocytes [43]. The effect of the dietary treatment of Iberian pigs on plasma leptin concentration has been reported by several authors [14,21]. To our knowledge, no studies have been previously carried out to determine plasma leptin levels in Iberian pigs fed in free-range conditions, according to the traditional system known as *montanera*, with acorns and pasture, which is recognized as a poor natural source of protein [44]. Therefore, there are no studies that compare the free-range and the intensive feeding systems. However, differences in the plasma leptin levels of Iberian pigs fed intensively with crude-protein diets or lys-deficient diets have been previously evidenced [14], with the pigs with greater relative weights of fatty components (backfat, belly, kidney fat, and mesenteric fat) [14] and intramuscular fat [15] demonstrating higher hormone levels. Nevertheless, these authors revealed only slight differences in subcutaneous fat depth measured at the first and last rib and last lumbar vertebra [15], in agreement with our results, where no differences in backfat thickness between the three feeding systems studied were detected. Furthermore, in accordance with the study carried out previously in our lab, Iberian pigs fed according to the FR system, with higher plasma leptin concentrations, exhibited greater intramuscular fat content compared to pigs fed on diets with higher protein contents (SP) [19].

Plasma insulin concentration was also influenced by diet, with SP pigs demonstrating higher levels than FR and LP pigs (*p* < 0.01). Some studies reported that the reduction in the crude-protein [45] or lysine [46] content in the pig diets resulted in a decrease in plasma insulin levels, which is in agreement with our results. The difference between diets in plasma insulin concentration, in spite of a higher intake of carbohydrates in LP compared to SP pigs, could be attributable to the higher protein content in SP diet, since the ingestion of protein elicits a rise in insulin in non-ruminants [47]. On the contrary, other studies reported no difference in the plasma insulin concentration of pigs fed different levels of protein [21,48]. These differences could be explained by the level of protein or lysine reduction in diets. Therefore, as suggested by Fernández-Figares et al. [21], apparently only severe protein restriction is able to elicit changes in the hormonal profile of pigs. The lipogenic action of insulin on adipose tissue is well documented [49]. The lack of differences in backfat thickness between Iberian pigs fed on the three diet treatments in our study could indicate that the lipogenic effect of insulin, as occurs with lipogenic enzyme activities [23], could be expressed less readily in subcutaneous fat than in muscle.

### 4.3. Effect of Diet and Adipose Tissue on Lipogenic Enzyme Activity

Lipogenic pathways could play a determining role in the amount of lipid deposited in tissues [15], because more than 80% of the total fatty acid deposition in pig tissues is attributed to de novo synthesis [50]. To further investigate adipogenesis and metabolism in different carcass adipose tissues, such as backfat—total (TB) and the SC1, SC2, and SC3 layers—and perirenal (PE) fat in Iberian pigs fattened in free-range and in intensive conditions, with different protein contents in their diets, the activity of four lipogenic enzymes (G6PDH, ME, G3PDH, and FAS) was analyzed. These enzymes play an important role in lipid metabolism: G6PDH and ME are involved in reduced nicotinamide adenine dinucleotide phosphate (NADPH) supply for de novo fatty acid synthesis; G3PDH produces glicerol-3-phosphate, involved in triglyceride synthesis, from the glucose; and FAS catalyzes the palmitate synthesis pathway from malonil-CoA and is therefore also involved in de novo fatty acid synthesis.

Differences in lipogenic enzyme activity between lean and fatty pig breeds during the fattening period have been pointed out by several authors [51,52], which lead to a more intense lipid metabolism [15] and greater capacity for tissue lipid synthesis in fatty pigs, such as Iberian pigs, as compared to conventional pigs [53]. The effect of animal tissue, muscle, and adipose tissue on lipogenic enzyme activity is also well known [13,23], which would suggest different regulatory lipid metabolism mechanisms for subcutaneous and intramuscular fat [15]. Finally, feeding regimes involving dietary crude protein [22,23] or lysine [13,15] restriction can also affect the lipogenic enzyme activity in pigs, although this effect is closely related to the fatty tissue studied, as mentioned above. The results of lipogenic enzyme activity in adipose tissues between diet regimes in our study showed a significant difference between Iberian pigs fed on natural resources in free-range conditions (FR), with lower G6PDH, ME, and G3PDH activities, compared to pigs fed intensively on concentrates (LP and SP). A possible explanation for this difference between the FR group and intensive fed pigs could be the very high amount of feed consumed by the SP and LP pigs (see Tejeda et al. [19]), which results in high-energy intake, leading to an increase in fat synthesis and consequently an increase in lipogenic enzyme activity. However, no differences in backfat thickness measured at the end of the fattening period were detected, probably because the FR pigs exhibited a lower average daily intake and needed more days to reach slaughtering weight compared to the LP and SP pigs (56.8 vs. 45.4 and 46.3 days, respectively) [19]. In contrast, the present results evidenced that diet protein restriction in Iberian pigs fed intensively with standard (SP) and low-protein concentrates (LP) had no effect on backfat and perirenal fat lipogenic enzyme activity, which is in agreement with previous studies carried out with protein [51] or lysine [11] deficient diets, both in Iberian and lean pigs, which could help explain the absence of differences in the carcass traits of the pigs, specifically the backfat thickness and thickness increase in the three backfat layers studied. The differences in the enzymatic activity of lipogenic enzymes between subcutaneous fat and muscle, as suggested by Palma Granados et al. [15], could be the cause of the significant effect of low-protein diet on IMF content in Iberian pigs previously reported in our lab [19], and by other authors [54], compared to the absence of effect on subcutaneous or perirenal fat. Nevertheless, it would be interesting to study further the lipogenic enzyme activity in Iberian pig muscles to find more evidence of the effect of low-protein diets on carcass and meat quality.

With respect to the effect of adipose tissue on the lipogenic enzyme activity, our results show differences between backfat (TB, SC1, SC2, and SC3) and PE fat from Iberian pigs, with the metabolic enzyme activity being less intense in global terms in PE compared to subcutaneous fat. These results confirm the effect of tissue on lipogenic enzyme activity as studies comparing muscle and subcutaneous tissues demonstrated [13,15,23]. This fact could help explain the significantly different enzyme activity described in our study in PE with respect to TB, SC1, SC2, and SC3, since PE is a tissue that develops at a later age compared to other fat depots [35].

### 4.4. Effect of Diet and Adipose Tissue on Fatty Acid Composition

The fatty acid composition of pig tissues can be altered by several factors, such as genotype, sex, age, slaughter weight, and environmental temperature, with the diet being the main factor through which this fatty acid profile can be modified [54,55]. In the present study, no effect was observed regarding protein restriction in the diet in the different adipose tissues studied. In agreement with our results, Daza et al. [56] reported that a low-crude-protein diet had no effect on the major fatty acid proportions of the total subcutaneous fat. In addition, partially agreeing with our results, Aquilani et al. [10] did not detect any effect as regards crude-protein restriction on the backfat inner layer from Cinta Senese pigs, corresponding to the middle layer in our study. However, this was opposite to the findings of most studies, which reported dietary crude-protein reduction having a significant effect on the fatty acid composition of subcutaneous [23,38] and intramuscular [38] fat from finishing lean pigs, as well as in Iberian piglets growing from 10 to 25 kg in body weight and fed on low-lysine diets [14,15]. All these studies have an increase in SFA and MUFA, and a reduction in PUFA in pigs fed on low-protein compared to standard-protein diets in common, which could indicate an activation of lipogenic enzymes at the adipose or muscular tissue [57], and consequently, the increase in de novo fatty acid synthesis and stearoyl-CoA desaturase activity [23]. In this regard, it has been pointed out that these discrepancies in the results described in the literature could be related to the level of dietary protein reduction [38], in addition to genetic factors and the slaughter age of the pigs in the different studies [14,15,38]. The increase in de novo synthesis in pigs fed on the LP diet compared to those fed the SP diet could be diminished as a result of the high proportion of oleic acid (about 72%) in the concentrated feed used in our study. Moreover, it is important to highlight again that the Iberian pig is a rustic fatty pig slaughtered at high weights, which could also contribute to the lack of effect related to diet protein level on fatty acid composition of the different fatty tissues studied. The dietary effect detected in the current study was associated with free-range rearing vs. feeding in intensive conditions, since FR pigs exhibited lower C16:0, C18:0, C20:0, and SFA, and higher C18:1, C18:2, C18:3, MUFA, and PUFA in adipose tissues compared to LP and SP pigs. An explanation for these results could be related to the composition of natural resources (acorn and grass) consumed by pigs during the fattening period [19,58,59].

Regarding the effect of carcass fat tissue on the fatty acid composition, our results are in accordance with those previously reported [60] for Iberian pigs, with differences between the three subcutaneous layers studied, with the outer layer (SC1) showing the highest proportions of MUFA, mainly C18:1 n-9, and PUFA, mainly C18:2 n-6 and C18:3 n-3, and the lowest of SFA, mainly C16:0 and C18:0. The same trend was observed in Cinta Senese pigs, an obese genotype characterized by great lipogenic potential [10], similar to the Iberian pig, although these authors only studied two subcutaneous fat layers. These differences in the fatty acid composition could be attributed to variations in lipogenic enzyme activity in the different fat depots. The hierarchy observed in the G3PDH activity, with higher values in the outer compared to the inner layers, and the observed decreasing tendency in G6PDH and ME activities in the outer, middle, and inner backfat layers (SC1, SC2, and SC3, respectively) and in PE tissue in our study, evidenced the tissue-specific lipogenic enzyme activity.

## 5. Conclusions

Under the experimental conditions of our study, which was carried out on Iberian pigs, a fatty, rustic breed, slaughtered at an advanced age and weight, dietary crude-protein restriction during the final fattening period prior to slaughter did not affect the carcass fatness or the enzyme activity and fatty acid composition of fatty tissues. This fact provides relevant details regarding the usefulness of feeding Iberian pigs with LP diets in intensive conditions to improve meat quality. On the other hand, our results showed that the free-range feeding system, characterized by a low-crude-protein content, increased the plasma leptin content, but had no effect on carcass fatness traits, supporting the described leptin resistance of the Iberian pig. The extensive feeding system, which implies a slower pig growth, and consequently, a longer fattening period in order to reach the final fattening weight as compared to intensive feeding system, decreased the lipogenic enzyme activity and modified the fatty acid composition of fatty tissues. Additionally, the present results confirm that lipogenic metabolic pathways are adipose tissue-specific, with the different adipose tissues being affected by the feeding system.

## Figures and Tables

**Table 1 animals-10-01917-t001:** Carcass traits and plasma hormone concentrations from Iberian pigs fed with the experimental diets.

Items	FR	LP	SP	SEM	*p*-Value
Carcass traits					
Trimmed ham weight (kg)	11.4	11.4	11.6	0.10	0.772
Trimmed shoulder weight (kg)	7.4	7.5	7.9	0.10	0.068
Trimmed ham yield (%)	16.6	16.6	17.1	0.18	0.415
Trimmed shoulder yield (%)	10.8 ^a^	10.8 ^a^	11.7 ^b^	0.16	0.024
Backfat thickness (last rib) (mm) ^1^	62.8	62.6	58.0	1.86	0.490
Backfat thickness increasing (mm) ^2^					
SC1	0.7	0.4	0.5	0.05	0.129
SC2	2.1	2.6	2.8	0.25	0.577
SC3	1.6	2.0	2.2	0.18	0.358
Hormone concentration					
Leptin (ng/mL)	30.3 ^a^	21.2 ^b^	13.9 ^b^	1.97	0.001
Insulin (μU/mL)	10.4 ^a,b^	7.9 ^a^	11.8 ^b^	0.55	0.007

^a,b^ Values within a row with different superscripts differ significantly at *p* < 0.05. SEM, standard error of the mean. SC1, outer backfat layer. SC2, middle backfat layer. SC3, inner backfat layer. Diets: FR, Iberian pigs reared in free-range conditions; LP, Iberian pig fed on experimental low-protein diet; SP, Iberian pig fed on experimental standard protein diet. ^1^ Measurements were carried out ultrasonically, 24 h before slaughter, 10 cm from the dorsal midline at the level of the last rib. ^2^ Increase was calculated as the difference between the two measurements carried out ultrasonically on the first and the last day (24 h before slaughter) of the fattening period, and divided by the total number of days in this period and multiplied by 10 (backfat thickness increasing every 10 days).

**Table 2 animals-10-01917-t002:** Effect of diet and adipose tissue on the lipogenic enzyme activities of backfat and perirenal fat from Iberian pigs. Results are expressed in nmol of nicotinamide adenine dinucleotide phosphate (NADPH) produced (ME, G6PDH) or consumed (FAS), and NADH consumed (G3PDH) per min and per g lipids.

	Diet			Tissue						*p*-Value		
	FR	LP	SP	TB	SC1	SC2	SC3	PE	SEM	Diet	Tissue	Int.
G6PDH	352 ^a^	456 ^b^	472 ^b^	497 ^a^	484 ^a,b^	438 ^a,b^	423 ^b^	297 ^c^	12.9	0.000	0.000	0.851
ME	315 ^a^	490 ^b^	459 ^b^	357 ^a^	481 ^b^	477 ^b^	413 ^a,b^	379 ^a^	15.4	0.000	0.000	0.458
G3PDH	140 ^a^	213 ^b^	212 ^b^	238 ^a^	207 ^a,b^	186 ^b,c^	155 ^c^	158 ^c^	6.6	0.000	0.000	0.993
FAS	29	34	33	34	37	28	29	36	1.7	0.689	0.434	0.951

^a,b,c^ Values within a row with different superscripts differ significantly at *p* < 0.05. SEM, standard error of the mean. Int., interaction Diet × Tissue. G6PDH, glucose-6-phosphate dehydrogenase; ME, malic enzyme; G3PDH, glycerol-3-phosphate dehydrogenase; FAS, fatty acid synthase. Diets: FR, Iberian pigs reared in free-range conditions; LP, Iberian pig fed on experimental low-protein diet; SP, Iberian pig fed on experimental standard protein diet. Tissues: TB: backfat, at the level of the tailbone; SC1, outer backfat layer; SC2, middle backfat layer; SC3, inner backfat layer; PE, perirenal fat.

**Table 3 animals-10-01917-t003:** Effect of diet and adipose tissue on fatty acid composition (%) of the backfat and perirenal fat from Iberian pigs.

	Diet			Tissue						*p*-Value		
	FR	LP	SP	TB	SC1	SC2	SC3	PE	SEM	Diet	Tissue	Int.
C14:0	1.21	1.21	1.18	1.21	1.21	1.20	1.16	1.21	0.010	0.335	0.402	0.082
C16:0	20.79 ^a^	21.61 ^b^	21.55 ^b^	20.29 ^a^	20.02 _a_	21.03 ^b^	21.57 ^b^	23.66 ^c^	0.144	0.000	0.000	0.053
C16:1 n-7	2.04	2.14	2.22	2.22 ^a^	2.34 ^a^	2.26 ^a^	2.03 ^a,b^	1.81 ^b^	0.039	0.116	0.000	0.635
C17:0	0.34 ^a^	0.31 ^b^	0.34 ^a^	0.28 ^a^	0.33 ^a,b^	0.39 ^b^	0.33 ^a^	0.32 ^a^	0.007	0.044	0.000	0.969
C17:1	0.34	0.32	0.35	0.33 ^b^	0.38 ^a,b^	0.41 ^a^	0.33 ^b^	0.24 ^c^	0.008	0.273	0.000	0.993
C18:0	10.66 ^a^	12.14 ^b^	11.92 ^b^	9.63 ^a^	9.51 ^a^	11.12 ^b^	12.22 ^c^	15.39 ^d^	0.237	0.000	0.000	0.863
C18:1 n-9	51.10 ^a^	49.88 ^b^	49.99 ^b^	53.04 ^a^	52.72 ^a^	50.07 ^b^	50.31 ^b^	45.49 ^c^	0.293	0.003	0.000	0.880
C18:2 n-6	9.93 ^a^	8.70 ^b^	8.84 ^b^	9.10 ^a,b^	9.53 ^a,b^	9.66 ^b^	8.62 ^a^	8.88 ^a,b^	0.120	0.000	0.013	0.813
C18:3 n-3	0.80 ^a^	0.71 ^b^	0.71 ^b^	0.69 ^a,b^	0.74 ^a,b,c^	0.80 ^b,c^	0.66 ^a^	0.81 ^c^	0.014	0.012	0.002	0.999
C20:0	0.18 ^a^	0.21 ^a,b^	0.25 ^b^	0.19	0.25	0.22	0.21	0.19	0.010	0.042	0.429	0.533
C20:1 n-9	1.47 ^a^	1.67 ^b^	1.58 ^a,b^	1.82 ^a^	1.70 ^a,b^	1.60 ^b^	1.55 ^b^	1.19 ^c^	0.028	0.001	0.000	0.342
C20:2 n-9	0.63	0.62	0.59	0.70 ^a^	0.69 ^a^	0.68 ^a^	0.55 ^b^	0.43 ^c^	0.013	0.135	0.000	0.172
C20:3 n-6	0.10	0.09	0.10	0.09 ^a,b^	0.11 ^a^	0.11 ^a^	0.09 ^a^	0.07 ^b^	0.002	0.314	0.000	0.446
C20:4 n-6	0.15	0.13	0.15	0.13 ^a^	0.15 ^a,b^	0.17 ^b^	0.14 ^a,b^	0.14 ^a,b^	0.003	0.138	0.012	0.908
C20:3 n-3	0.26	0.26	0.25	0.27 ^a,b^	0.31 ^a^	0.30 ^a^	0.23 ^b^	0.16 ^c^	0.007	0.759	0.000	0.474
SFA	33.18 ^a^	35.47 ^b^	35.23 ^b^	31.60 ^a^	31.32 ^a^	33.95 ^b^	35.48 ^c^	40.78 ^d^	0.368	0.000	0.000	0.583
MUFA	54.95 ^a^	54.02 ^b^	54.14 ^b^	57.42 ^a^	57.14 ^a^	54.33 ^b^	54.23 ^b^	48.74 ^c^	0.325	0.036	0.000	0.929
PUFA	11.86 ^a^	10.51 ^b^	10.63 ^b^	10.97 ^a,b,c^	11.53 ^a,b^	11.72 ^a^	10.29 ^c^	10.49 ^b,c^	0.146	0.000	0.002	0.856

^a,b,c^ Values within a row with different superscripts differ significantly at *p* < 0.05. SEM, standard error of the mean. Int., interaction Diet × Tissue. SFA, total saturated fatty acids; MUFA, total monounsaturated fatty acids; PUFA, total polyunsaturated fatty acids. Diets: FR, Iberian pigs reared in free-range conditions; LP, Iberian pig fed on experimental low-protein diet; SP, Iberian pig fed on experimental standard protein diet. Tissues: TB: backfat, at the level of the tailbone; SC1, outer backfat layer; SC2, middle backfat layer; SC3, inner backfat layer; PE, perirenal fat. Results are expressed as means in percentage of a total of 15 fatty acids identified.

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
