# Peer review of "Free-Range and Low-Protein Concentrated Diets in Iberian Pigs: Effect on Plasma Insulin and Leptin Concentration, Lipogenic Enzyme Activity, and Fatty Acid Composition of Adipose Tissue"

_animals, 2020, doi:10.3390/ani10101917_

Round 1

Reviewer 1 Report

The present article deals with relatively interesting topic of studying several aspects of lipid metabolism in relation to nutrition which is specific in rustic fatty pig breeds. The article, however, being a continuation of the previously published work (Tejeda et al., 2020, still focuses a lot on the concept of intramuscular fat (i.e. its increase in the case of low protein diet), which was not even analysed as a part of present manuscript (either in terms of fatty acid composition or lipogenic enzyme activities). The article should therefore be thoroughly rewritten in order to focus solely on the investigated/presented traits.

Specific comments:

Lines 21-25: Rather unclear, please simplify/rewrite.

Line: 22: in all three groups

Line 23: despite the level of hormones

Line 30: twenty four castrated male Iberian pigs (eight per feeding diet) were

Lines 38-42: Please specify the fat depots.

Line 96: The fattening period was adapted to similar slaughter weight, please mention this informaiton in the text.

Lines 187-192: This part is not dealing with the effect of treatment group, in addition, similar (highly related) traits (ham and shoulder weights) have already been reported in a previously published part of the reseach.  Removal or substantial shortening of this part is therefore suggested.

Lines 188-190: The same numbers are already reported in the tables, please omitt (valid for entire text).

Line 214: Why multiplying the values by 10?

Lines 216-222: Given so much emphasis on the intramuscular fat, whay was this depot not studied in regard to the lipogenic enzyme activities?

Line 223: »total backfat« is it meant TB?

Lines 272-292: This part contains to much discussion that is not really related to the traits investigated in the study (i.e. IMF) and explaining the study aims (which should be in the introduction part). Extensive shortening of this part of the text is suggested.

Line 228: »only a layer« - which layer?

Lines 311-314: Sentence difficult to understand, please rewrite.

 Line 316: You probably mean intermediate fat layer, as there are 3 layers of fat (especially when dealing with fatty breeds, where third-internal fat layer is developed).  

 Line 318: what is meant by »lowest scores«?  The lowest growth rate or the lowest amount?

Lines 320-333: This part should be significantly shortened, as you have not shown any differences in fatness.

Lines 354-357, 370-372, 410-415, 422-425: Similar comment, IMF was not a part of this study and despite this it is extensively discussed.

Lines 361-362: Please rewrite. The primary reason for increasing insulin levels is carbohydrates, which, on the opposite are most probably increased in low-protein diets. In addition, the insulin levels change in response to the feed ingestion, whereas you report 12h fasting time before the slaughter.

Lines 399-402: Animal age may also directly affect lipogenic capacity.

Lines 430-441: Again too much discussion not related to the present results.

Line 448: Third or second layer?

Reviewer 2 Report

Review Report

  • A summary

The study was aimed at investigating the effects of different concentrated diets based on free-range and low-protein on the metabolism, development, and composition of the adipose tissue of Iberian pigs. The approach was in-vivo feeding experiment and backfat measures, some carcass measurements, plasma insulin and leptin concentration, lipogenic enzyme activity, and fatty acid composition of adipose tissue have been used as variables. There are cited to a previous study that experimental design and other performance parameters are reported.

  • Broad comments

The experiments are logical, seemed to be carefully done and the paper is thoroughly written. However, some reformulations are needed in the materials and methods and discussion parts. I would suggest that the results be discussed more systematically. The methods should be more adequately described, and the limitations of the study design should be mentioned in the discussion part.

  • Specific comments

Page 1 line 36: glucose-6-phospahe dehydrogenase should be changed to glucose-6-phosphate dehydrogenase.

Page 3 line 96: There were different fattening periods for different feeding groups. (45.4, 46.3, and 56.8 days for 96 LP, SP, and FR pigs, respectively). How can this affect the interpretation of the results and the conclusion?

Page 3 line 117: Did you obtain the blood samples before the fattening period?

In order to have a better understanding of the changes in plasma hormones, it is important to compare the values at the ending of an experiment with the basal values. It is therefore recommended that the animals are kept some days before the start of experiment in an acclimatization period where each individual can be adapted to experimental conditions.

Page 5 line 191: (Tejeda et al., 2020) should be replaced with the number of the corresponding reference! (#19?)

Page 6 line 232: Which statistical model did you use to study the interaction between diet and tissue? It should then be mentioned in materials and methods!

Page 7 line 262: Which statistical model did you use to study the interaction between diet and tissue? It should then be mentioned in materials and methods!

Page 7 lines 281-283: I would suggest reformulation of this phrase. This is a bit misleading if you are referring to the previous study while writing about “the purpose of this current study”.

Page 8 line 302: Based on the information from your previous study, the FR pigs’ diets contained both acorn and grass. Although you mentioned that acorns were the main component of FR diets, the final composition of the FR diets is not clearly described since it is based on the traditional way where pigs are fed on natural resources. In this study design, the feed intake was not recorded! How this form of feeding method in a feeding experiment where two other types of more controlled diets are included and are compared at the same time can affect interpretation of the results is an important source for discussion!

Page 8 line 334: Please see the previous comment on page 3 line 117 about blood collection.

Round 2

Reviewer 1 Report

In the revised version, the authors have sufficiently addressed all raised issues, therefore I can recommend the manuscript for publication.